# Risk of New-Onset Dementia in Patients with Chronic Kidney Disease on Statin Users: A Population-Based Cohort Study

**DOI:** 10.3390/biomedicines11041073

**Published:** 2023-04-02

**Authors:** Gwo-Ping Jong, Tsung-Kun Lin, Jing-Yang Huang, Pei-Lun Liao, Tsung-Yuan Yang, Lung-Fa Pan

**Affiliations:** 1Division of Cardiology, Department of Internal Medicine, Chung Shan Medical University Hospital, Taichung 40201, Taiwan; 2Institute of Medicine, College of Medicine, Chung Shan Medical University, Taichung 40201, Taiwan; 3School of Pharmacy, National Defense Medical Center, Taipei 114201, Taiwan; 4Department of Medical Research, Chung Shan Medical University Hospital, Taichung 40201, Taiwan; 5Department of Cardiology, Taichung Armed Forces General Hospital, Taichung 41168, Taiwan; 6Department of Medical Imaging and Radiological Science, Central Taiwan University of Science and Technology, Takun, Taichung 406053, Taiwan

**Keywords:** new-onset dementia, statin, chronic kidney disease

## Abstract

Patients with chronic kidney disease (CKD) are at a higher risk for developing dementia than the general population. Clinical studies have investigated the effects of statin use on new-onset dementia (NOD) in patients with CKD; however, the findings are inconsistent. This study examines the association between the use of statins and NOD in patients with CKD. We conducted a nationwide retrospective cohort study using the Taiwan Health Insurance Review and Assessment Service database (2003–2016). The primary outcome assessed the risk of incident dementia by estimating the hazard ratios and 95% confidence intervals. Therefore, multiple Cox regression models were conducted to analyse the association between statin use and NOD in patients with CKD. There were 24,090 participants with statin use and 28,049 participants without statin use in patients with new-diagnosed CKD; the NOD event was 1390 and 1608, respectively. There was a trend of reduction association between statin users and NOD events after adjusted sex, age, comorbidities, and concurrent medication (adjusted HR 0.93, 95% CI 0.87 to 1.00) in the 14 years of the follow-up. Sensitivity test for the propensity score 1:1 matched analyses showed similar results (adjusted HR 0.91, 95% CI 0.81 to 1.02). The subgroup analysis also identified the use of statins as having a trend against developing NOD in patients with hypertension. In conclusion, statin therapy may effectively reduce the risk of NOD in patients with CKD. More studies are needed to credibly evaluate the effects of statin therapy on the prevention of NOD in patients with CKD.

## 1. Introduction

Dementia and chronic kidney disease (CKD) are major world health concerns, but their association is rarely considered [1,2]. Because of the aging population and improved survival after CKD, more patients are living with these two comorbidities [1,2,3]. Older patients with CKD are more likely to suffer from dementia. The incidence of dementia was much higher in patients with CKD than in the general population [4]. Dyslipidemia caused by renal dysfunction is the most common complication in patients with CKD, contributing further renal damage and renal function deterioration [5,6]. Dyslipidemia and CKD are associated with an increased risk of dementia [7,8]. Therefore, lipid regulation therapy in patients with CKD is quite important and well-accepted [9].

Statins (inhibitors of 3-hydroxy-3-methylglutaryl coenzyme A reductase) are an important component of the medical management of patients with dyslipidemia [9,10,11]. Furthermore, statins reduce cardiovascular morbidity in patients with CKD, including those without established coronary heart disease [11]. However, previous randomized clinical trials have shown eighty-one percent of trials excluded patients with severe CKD and seventy-six excluded those with moderate CKD [12]. Therefore, the results of previous studies regarding the effects of statin use on new-onset dementia (NOD) in patients with CKD are uncertain and worthy of examination [13]. In this study we aimed to evaluate the risk of NOD associated with statin use in a nationwide cohort study of Taiwanese patients with CKD.

## 2. Materials and Methods

### 2.1. Data Source

This retrospective cohort study examined a population-based cohort using insurance claims data provided by the Taiwanese Bureau of National Health Insurance from January 2003 to December 2016. This database contains anonymized patient numbers, sex, age, three diagnostic codes for outpatient visits and five for inpatient visits, medications, drug doses, medical expenditures, and hospital and physician information. The prescription table contains the quantity and expenditure for all the administered drugs, operations, and treatments.

Patients aged at least 40 years with a recorded new diagnosis of CKD with or without statins between 2005 and 2016 were included. Statin users were patients who received at least one statin prescription for 180 days during the study period, whereas non-statin users did not receive a statin prescription throughout the study.

### 2.2. Patient Selection

In this retrospective cohort study, we performed a population-based cohort study using insurance claims data provided by the Taiwanese Bureau of National Health Insurance from January 2003 to December 2016. This database contains anonymized patient numbers, sex, age, three diagnostic codes for outpatient visits and five diagnostic codes for inpatient visits, medications, drug doses, medical expenditures, and hospital and physician information. The prescription table contains the quantity and expenditure for all the administered drugs, operations, and treatments.

The study population comprised patients with CKD (ICD-9-CM: 585; ICD-10-CM: N18) admitted to the hospital or outpatients between 2003 and 2016. The list of ICD-9 and ICD-10 codes used to define the inclusion of patients with CKD, study events, and comorbidities is presented in Appendix A. At least one of the following enrolment criteria needed to be met for inclusion in this study: (1) two or more outpatient visits within a year, (2) all statins were continuously prescribed to the patients for >6 months during the follow-up, or (3) one or more inpatient admissions for CKD. Patients with (1) a history of dementia before 2005, (2) dialysis patients, or (3) age <40 years were excluded from the study. Finally, the study group comprised 24,090 patients with CKD who were statin users, and the control group included 28,049 randomly selected patients with CKD who were not statin users (Figure 1). Propensity score matching was also applied with age, sex, comorbidities, and drug index date at a ratio of 1:1 for sensitivity analysis in patients with CKD who were statin and non-statin users (Figure 1).

### 2.3. Study Outcome and Variables

The primary endpoint was NOD, which was defined by the time a dementia code first appeared in the inpatient or outpatient claim records during the study period. Dementia-related comorbidities were defined according to the ICD-9-CM code and ICD-10-CM code (Appendix A). Patient demographic characteristics were assessed on the index date. The demographic variables included: gender, age, diabetes duration, comorbidities (coronary artery disease, hypertension, heart failure, peripheral artery disease, ischemic stroke, diabetes mellitus, chronic obstructive lung disease, and peptic ulcer) and concurrent medication (aspirin, warfarin, clopidogrel, diuretics, beta-blockers, calcium channel blockers, angiotensin converting enzyme inhibitors, angiotensin II receptor antagonists, insulin, and oral hypoglycemic agents). Comorbidities closest to the index date, and medication use, were assessed during a 180-day baseline period.

### 2.4. Statistical Analysis

Data are presented as valid percentages and mean values with a standard deviation. Differences in demographic data and clinical characteristics between statin users and non-statin users were examined using the t-test for continuous variables, whereas chi-square tests were used for categorical variables. The Cox proportional hazard regression model was used to compare the risk of NOD development between statin users and non-statin users. Adjusted hazard ratios (HRs) and 95% confidence intervals (CIs) were calculated, with the adjustment for age, sex, concurrent medication, and comorbidities at baseline for developing study events. The risk of study outcomes over time for statin users compared with non-stain users was determined using survival analysis by the Kaplan–Meier method.

We also conducted a sensitivity analysis to test the robustness of our primary findings. Initially, a propensity score was calculated for each patient to minimize confounding by indication, such as other risk factors existing between statin users and non-statin users. Then, propensity score matching (1:1) and absolute standardized differences (ASD) were calculated to estimate the difference between the two groups. An ASD <0.10 suggested a negligible difference in potential confounders between the two groups.

We conducted subgroup analyses stratified by sex, age, comorbidities, and concurrent medications at baseline for the primary outcomes of NOD. Finally, we conducted a subgroup analysis stratified by CKD for the interaction effect between CKD and statin on the risk of dementia. Significance was considered at *p* < 0.05. All statistical calculations were performed using Statistical Analysis Software, version 9.3 (SAS Institute, Inc., Cary, NC, USA).

## 3. Results

### 3.1. Demographic Clinical Characteristics of the Study Population

A total of 52,139 patients with CKD were enrolled in this study, which included 24,090 statin users and 28,049 non-statin users from 2005 to 2016 (Figure 1). The median age of the cohort was 65 years, and the median duration of follow-up was 61 months. There were more male patients (36,950) than female patients (15,189). At baseline, statin users had more comorbidities, except for chronic obstructive lung disease and peptic ulcer, and used more concurrent medication than non-statin users (Table 1).

### 3.2. Relative Risk of NOD

Statin users recorded 1390 primary NOD events, whereas non-statin users recorded 1608. A trend of lower risk was found between statin use and NOD events after adjustment for sex, age, comorbidities, and concurrent medication (adjusted HR 0.93, 95% CI 0.87–1.00) in the 12 years of the follow-up (Table 2). The survival analysis with the Kaplan–Meier method also demonstrated no significant association between statin users and non-statin users (*p* = 0.6577) (Figure 2).

### 3.3. Sensitivity Analysis of the Relative Risk of NOD in the Propensity Score Matching Analysis

Sensitivity analysis under 1:1 propensity score matching also showed a trend of decreasing risk of NOD among statin users compared with non-statin users (HR 0.92; 95% CI 0.84–1.03, *p* = 0.0673) in patients with CKD (Table 3). Similarly, a consistent association was found between statin users and NOD after adjustment for sex, age, comorbidities, and concurrent medication (adjusted HR [aHR] 0.91; 95% CI 0.81–1.02, *p* = 0.0598) compared to non-statin users.

### 3.4. Subgroup Analysis

We also performed subgroup analysis based on the study participants. Results from the subgroup analyses were partly consistent with the main analyses (Table 4). A similar finding was seen for participants with hypertension (aHR 1.02, 95% CI 0.92 to 1.14). The risk of NOD in female participants (aHR 0.87, 95% CI 0.80 to 0.94) is decreased compared to male participants among patients with statin use. Inversely, a higher risk of treatment effect of NOD was seen for participants aged ≥50 years than in participants aged < 50 years (aHR 1.42, 95% CI 1.18 to 1.69). Among patients with coronary artery disease, heart failure, peripheral artery disease, ischemic stroke, diabetes mellitus, chronic obstructive lung disease, or peptic ulcer receiving statins, the risk of NOD (aHR: 1.12, 95% CI: 1.03–1.21; aHR: 1.32, 95% CI: 1.11–1.43; aHR: 1.12, 95% CI: 1.03–1.21; aHR: 1.56, 95% CI: 1.43–1.69; aHR: 1.24, 95% CI: 1.15–1.34; aHR: 1.21, 95% CI: 1.11–1.31; or aHR: 1.11, 95% CI: 1.03–1.19; all *p* < 0.0001) was increased compared with non-statin users (Table 4). Similarly, we analysed these abovementioned risks in patients with concurrent medication. The use of statin in patients with concurrent prescription with aspirin, clopidogrel, diuretics, beta-blockers, calcium channel blockers, angiotensin converting enzyme inhibitors, angiotensin II receptor antagonists, and insulin was not associated with increased NOD for except warfarin and oral hypoglycemic agent prescription. Finally, we also subgroup analysis stratified by CKD, and the result shows no significant interaction effects between CKD and statin on the risk of NOD (Table 5).

## 4. Discussion

This study showed a trend of reduced association between statin use and NOD risk in patients with CKD. The decrease in risk of dementia in patients with CKD was greater in female patients. This study also demonstrated that in older adult patients (aged > 50 years), increased risk of incident dementia was found between statin users and non-statin users.

A recent prospective community-based cohort study reveals a significant association between CKD and dyslipidemia and high risk for development of NOD [8]. Although pathophysiological mechanisms of this association are largely unknown, certain hypotheses describe possible mechanisms [14,15,16]. Patients with CKD and dementia show elevated levels of endogenous inhibitors of NO synthesis and decreased nitric oxide (NO) metabolites [17]. Low NO may regulate microcirculation and blood–brain barrier. Another mechanism is via strained vessels, wherein vessels are exposed to very high pressure and maintain a high vascular tone. The brain and its vessels share similar anatomic and haemodynamic features. Brain vessels undergo similar mechanisms when responding to vascular strain and incur similar pressure-induced injuries [18]. Microvascular damage alters the hemodynamics of the neurovasculature that contributes to dementia [18]. Small vessel disease in the kidney may also indicate the presence of small vessel disease in the brain [18,19,20,21]. A previous study showed stronger effects on vascular dementia in areas primarily affected by microvascular disease. It concluded that the association could be mediated by shared microvascular pathology in the kidney and the brain [22].

Previous studies have found that statins have pleiotropic effects in the modulation or prevention of endothelial dysfunction, inflammation, and oxidative stress [23]. The proposed mechanisms for the effects of statins on dementia remain unclear. It has been suggested that dementia is caused by large vessel disease and/or microvascular damage, and the risk factors for each are slightly different [24]. The most evident protective effect statins have on cognition is the prevention of stroke and possible subsequent vascular dementia. Statins may not only affect stroke risk but may also prevent microvascular infarcts that lead to dementia without an acute stroke. Statins have been associated with a reduced risk of all-cause dementia. It is possible that statins directly affect the pathological changes that lead from Alzheimer’s to dementia [24,25]. Statins inhibit 3-hydroxy-3-methylglutaryl-coenzyme A (HMG-CoA) reductase, resulting in a reduction in cholesterol synthesis. Polymorphisms in the HMG-CoA reductase gene change the risk, age of onset, and conversion of Alzheimer’s disease [26]. Studies have also shown a greater correlation between statins and a decreased risk of developing dementia in patients with congenital abnormalities leading to hyperlipidemia [27,28]. This could suggest that the positive effect of statins on long-term cognition is due to the restoration of cholesterol homeostasis in those who have congenital dyslipidemia. An optimal local cholesterol level may be necessary for normal physiologic processing. It is unlikely that decreasing peripheral serum cholesterol levels affect cognition; it is more probable that affecting cholesterol levels locally in the central nervous system is responsible for the cognitively impairing effects [29]. Previous reports suggest that high doses of lipophilic statins result in increased brain exposure to a medication [29], leading to the hypothesis that higher statin potency and lipophilicity may locally decrease cholesterol levels in the brain and eventually lead to cognitive impairment. However, a systematic review and meta-analysis showed no difference in the incidence of mild cognitive impairment between statin users and nonusers [30]. Additionally, the review found no significant effects or modifications of effects across all neurocognitive domains, regardless of whether the drug penetrates the blood–brain barrier or not, or of study duration, sample size, location, or cognitive health status. In our study, we observed that patients with CKD who were statin users had a trend of low risk for NOD. These independent effects have been shown in many studies with patients without CKD [31,32]. However, there is little information regarding the pleiotropic effects of statins in patients with CKD. Further studies will be needed to elucidate the benefits of statin use for NOD in patients with CKD.

Risk factors for dementia include old age and family history [33,34]. Old age causes atrophy of the cerebral cortex and hippocampus, which is associated with cognitive impairment and dementia. Previous epidemiological studies support the hypothesis that a high level of low-density lipoprotein (LDL) is associated with a higher risk for Alzheimer’s disease in the elderly [35]. Hence, statins may prevent Alzheimer’s disease, which leads to dementia, due to their role in cholesterol reduction. Low LDL cholesterol is also associated with a lower risk for dementia in the elderly. Cerebrovascular diseases have been suggested to contribute to neuropathological changes in cases of dementia [36]. A low level of LDL cholesterol has many vasoprotective functions and has been negatively correlated with cognitive decline in dementia patients [37,38]. In this study, patients aged > 50 years exhibited significantly increased risk for NOD, whereas among patients aged < 50 years, there was no significant association with risk of NOD development. This result indicates that age plays a major role in NOD development in patients with CKD. Further comprehensive research is warranted to elucidate the mechanisms underlying this association.

The strengths of our study include that it is population-based and has a large sample. Our findings were tested using propensity score matching to control for potential confounders, which made our hypothesis feasible. This study provides a comprehensive evaluation of the association between the use of statin and dementia in patients with CKD. We found a trend of reduced association between statin users and NOD events among patients with CKD, along with an overall 7% decreased risk as compared to non-statin users.

The results of this study should be interpreted with caution owing to certain limitations. First, all cases in this study were collected from the claimed data sets of the Taiwanese BNHI in which diagnoses were only based on physician reports; therefore, our findings have limited generalizability to patients in different areas of the world. Second, CKD was defined based on GFR; however, its diagnosis using ICD-9 or ICD-10 codes in this study may not reflect the course of renal function. Third, as this was an observational study, any cause-and-effect relationships between NOD and statins could not be determined. Hence, further randomized control trials are needed for more detailed findings. Fourth, information about data of estimated GFR or CKD stage, blood pressure, as well as total or LDL cholesterol levels was not available from the National Health Insurance Research Database. This is an important limitation in this study. However, because the data we used were population-based data, we assumed that there were no differences among the two groups. Further randomized clinical trials are needed to confirm our result. Finally, there are unmeasured or unknown confounders in our study; this may have resulted in residual confounding. Therefore, the analysis is affected by residual confounding in this study.

## 5. Conclusions

A trend of reduction in the association between statin use and NOD events was found after adjustment for sex, age, comorbidities, and concurrent medication. Further prospective randomized studies are needed to confirm this finding.

## Figures and Tables

**Figure 1 biomedicines-11-01073-f001:**
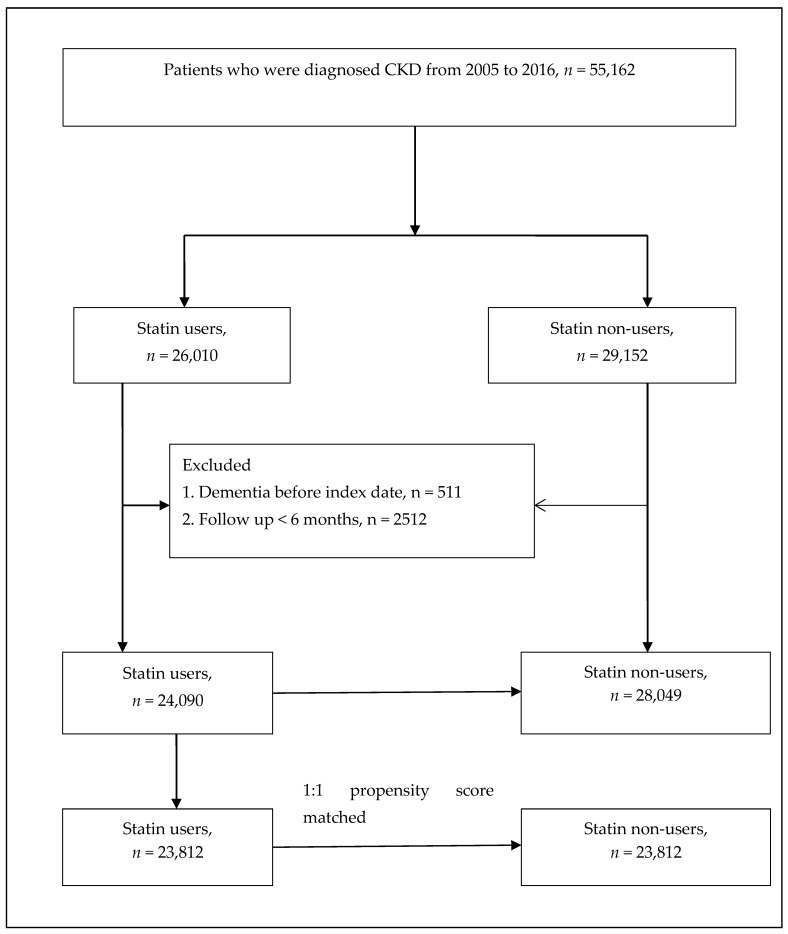
Patient flow chart.

**Figure 2 biomedicines-11-01073-f002:**
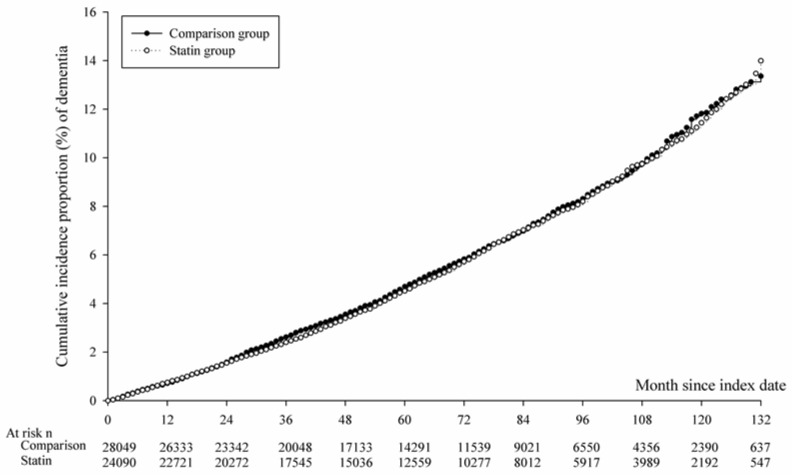
K-M curves of incidence probability of dementia in patients with chronic kidney disease, log rank *p* = 0.6577. Median of follow up time: Non-statin = 61 months, Statin user = 62.

**Table 1 biomedicines-11-01073-t001:** Baseline characteristics in patients with chronic kidney disease.

	Non-Statin	Statin User	*p*
N	28,049	24,090	
Sex			0.7891
Female	8185 (29.18%)	7004 (29.07%)	
Male	19,864 (70.82%)	17,086 (70.93%)	
Age group			0.0709
40–49	4004 (14.28%)	3470 (14.40%)	
50–59	9896 (35.28%)	8557 (35.52%)	
60–69	7886 (28.12%)	6796 (28.21%)	
70–79	4743 (16.91%)	4089 (16.97%)	
>=80	1520 (5.42%)	1178 (4.89%)	
Comorbidities			
Coronary artery disease	7332 (26.14%)	7966 (33.07%)	<0.0001
Hypertension	14,600 (52.05%)	15,850 (65.79%)	<0.0001
Heart failure	1058 (3.77%)	2897 (12.03%)	<0.0001
PAD	2822 (10.06%)	3573 (14.83%)	<0.0001
Ischemic stroke	3640 (12.98%)	4223 (17.53%)	<0.0001
Diabetes mellitus	7911 (28.20%)	9503 (39.45%)	<0.0001
COPD	5575 (19.88%)	4754 (19.73%)	0.6859
Peptic ulcer	13,959 (49.77%)	11,462 (47.58%)	<0.0001
Concurrent medication			
Aspirin	3264 (11.64%)	6104 (25.34%)	<0.0001
Warfarin	154 (0.55%)	251 (1.04%)	<0.0001
Clopidogrel	1264 (4.51%)	4169 (17.31%)	<0.0001
Diuretics	1032 (3.68%)	2853 (11.86%)	<0.0001
Beta-blockers	5811 (20.72%)	7494 (31.11%)	<0.0001
CCBs	7422 (26.46%)	9205 (38.21%)	<0.0001
ACEIs	2160 (7.70%)	3293 (13.67%)	<0.0001
ARBs	4459 (15.90%)	6874 (28.53%)	<0.0001
Insulin	182 (0.65%)	361 (1.50%)	<0.0001
OHAs	811 (2.89%)	9978 (41.42%)	<0.0001

ACEIs: angiotensin converting enzyme inhibitors; ARBs: angiotensin II receptor antagonists; CCBs: calcium channel blockers; COPD: chronic obstructive lung disease; OHAs: oral hypoglycemic agents; PAD: peripheral artery disease.

**Table 2 biomedicines-11-01073-t002:** The incidence rate of dementia in patients with chronic kidney disease.

	Non-Statin User	Statin User	*p*
N	28,049	24,090	
Follow up person-months	1,780,406	1,558,441	
Event of dementia	1608	1390	
Crude hazard ratio (95% CI)	Reference	0.98 (0.92–1.06)	0.0894
† Adjusted hazard ratio (95% CI)	Reference	0.93 (0.87–1.00)	0.0594

† adjusted hazard ratio, the covariates including sex, age, co-morbidities, and concurrent medication at baseline.

**Table 3 biomedicines-11-01073-t003:** Sensitivity analysis of incidence and risk of dementia after Propensity Score matching.

	Non-Statin User N = 23,812	Statin User N = 23,812	*p*
Follow up person-months	1,540,456	1,540,456	
Dementia cases	1415	1372	
‡ Crude HR (95% CI)	Reference	0.92 (0.84–1.03)	0.0673
† Adjusted HR (95% CI) †	Reference	0.91 (0.81–1.02)	0.0598

‡ crude incidence density rate (95% confidence interval) of dementia, per 1000 person-month. † adjusted hazard ratio, the covariates including sex, age, co-morbidities, and concurrent medication at baseline.

**Table 4 biomedicines-11-01073-t004:** Sub-group analysis, aHRs of dementia for use of statin.

	aHR	95% CI	*p*
Sex			
Female	0.87	0.80–0.94	0.0006
Male	Reference		
Age group			
40–49	Reference		
≥50	1.42	1.18–1.69	<0.0001
Comorbidities			
Coronary artery disease	1.12	1.03–1.21	0.0073
Hypertension	1.02	0.92–1.14	0.6599
Heart failure	1.32	1.11–1.43	<0.0001
PAD	1.12	1.03–1.21	0.0072
Ischemic stroke	1.56	1.43–1.69	<0.0001
Diabetes mellitus	1.24	1.15–1.34	<0.0001
COPD	1.21	1.11–1.31	<0.0001
Peptic ulcer	1.11	1.03–1.19	0.0061
Concurrent medication			
Aspirin	1.07	0.98–1.17	0.1110
Warfarin	1.43	1.08–1.90	0.0134
Clopidogrel	0.99	0.91–1.09	0.7677
Diuretics	1.02	0.93–1.12	0.6018
Beta-blockers	1.04	0.96–1.13	0.3657
CCBs	1.02	0.94–1.11	0.6106
ACEIs	1.01	0.92–1.12	0.8321
ARBs	0.99	0.90–1.08	0.7473
Insulin	1.01	0.92–1.13	0.8233
OHAs	1.11	1.02–1.20	0.0062

ACEI: angiotensin converting enzyme inhibitors; ARB: angiotensin II receptor antagonists; CCBs: calcium channel blocker; COPD: chronic obstructive lung disease; OHA: oral hypoglycemic agent; PAD: peripheral artery disease.; aHR: adjusted hazard ratio, the covariates include sex, age, co-morbidities, and concurrent medication at baseline.

**Table 5 biomedicines-11-01073-t005:** Sub-group analysis, aHRs of dementia for use of statin between CKD and without CKD.

Sub-Groups	aHR	95% CI	*p*
With CKD	0.931	0.865–1.003	0.0594
Without CKD	0.922	0.910–0.934	<0.0001
*p* for interaction			0.7953

## Data Availability

The datasets generated during and/or analysed during the current study are available from the corresponding author upon reasonable request.

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
