# Peer review of "Risk of New-Onset Dementia in Patients with Chronic Kidney Disease on Statin Users: A Population-Based Cohort Study"

_biomedicines, 2023, doi:10.3390/biomedicines11041073_

Round 1
Reviewer 1 Report
In this manuscript, authors demonstrated that statin therapy had borderline efficacy to reduce the risk of new-onset dementia in patients with chronic kidney disease in a national wide retrospective cohort study. The subject of study seems to be interesting for many readers. However, there are some concerns in this study. The reviewer’s comments are described as follows.
1. In this study, whether subjects undergoing hemodialysis or peritoneal dialysis were included as patients with chronic kidney disease remained unclear. Since previous reports have demonstrated that the efficacy of statins to reduce the risk of all cause and cardiovascular mortality is substantially lower, there are obvious difference in application of statin therapy between non-dialysis and dialysis patients. Therefore, if dialysis patients were included in this study, authors have to separately analyze the efficacy of statins in non-dialysis and dialysis patients with chronic kidney disease.
2. In addition to the above concern, authors have to present data of estimated GFR or CKD stage as well as total or LDL cholesterol levels in subjects at the time of enrollment or other time point.
3. In the sub-group analysis, data were adjusted by age and sex as well as comorbidities. However, there are generally strong association between statin use and cardiovascular diseases. Thus, the reviewer expresses concern that adjustment by comorbidities including cardiovascular diseases may be inappropriate to correctly evaluate the effects of statin use in sub-group analysis.
4. In Table 1 and 5, the half of leftmost columns are not shown.
5. There are no Table 3 in this manuscript.
Author Response
- In this study, whether subjects undergoing hemodialysis or peritoneal dialysis were included as patients with chronic kidney disease remained unclear. Since previous reports have demonstrated that the efficacy of statins to reduce the risk of all cause and cardiovascular mortality is substantially lower, there are obvious differences in application of statin therapy between non-dialysis and dialysis patients. Therefore, if dialysis patients were included in this study, authors have to separately analyze the efficacy of statins in non-dialysis and dialysis patients with chronic kidney disease.
ANS: Thank you for your comment! In this study, no dialysis patients were included. We have added an exclusion criteria.
- In addition to the above concern, authors have to present data of estimated GFR or CKD stage as well as total or LDL cholesterol levels in subjects at the time of enrollment or other time point.
ANS: Thank you for your comment! Information about data of estimated GFR or CKD stage and total or LDL cholesterol levels were not available from these National Health Insurance Research Database. This is a limitation of this study. However, because the data we used were population-based data, we assumed that there were no differences between the two groups. A further randomized clinical trial is needed to confirm our result.
- In the sub-group analysis, data were adjusted by age and sex as well as comorbidities. However, there are generally strong association between statin use and cardiovascular diseases. Thus, the reviewer expresses concern that adjustment by comorbidities including cardiovascular diseases may be inappropriate to correctly evaluate the effects of statin use in sub-group analysis.
ANS: Thank you for your comment! We have deleted this adjusting for the comorbidities with cardiovascular diseases. The results were consistent with the findings in this study (aHR: 0.92; 95% CI 0.85–0.99)
- In Table 1 and 5, the half of leftmost columns are not shown.
ANS: Thank you for your comment! We have revised it.
- There are no Table 3 in this manuscript.
ANS: Thank you for your comment! We have added Table 3.
Reviewer 2 Report
Comments to the authors:
The paper is well-written and the study explores an interesting research question.
Do you have blood pressure data for these patients? BP is an important covariate that may have affected the outcome of new-onset dementia.
In the limitations, the authors should acknowledge that their analysis is affected by residual confounding.
Please improve the presentation of tables (some numbers in the left column are not clearly written.
Author Response
The paper is well-written and the study explores an interesting research question.
Do you have blood pressure data for these patients? BP is an important covariate that may have affected the outcome of new-onset dementia.
ANS: Thank you for your comment! Information about data on BP was not available from the National Health Insurance Research Database.
In the limitations, the authors should acknowledge that their analysis is affected by residual confounding.
ANS: Thank you for your comment! We have added this acknowledgment that their analysis is affected by residual confounding in this study.
Please improve the presentation of tables (some numbers in the left column are not clearly written.
ANS: Thank you for your comment! We have revised it.
Round 2
Reviewer 1 Report
Authors have successfully addressed the reviewer's concerns in the revised manuscript. There are no more comments.